# Effects of Technology Use on Ageing in Place: The iZi Pilots

**DOI:** 10.3390/ijerph17145052

**Published:** 2020-07-14

**Authors:** Helen A.M. Silvius, Erwin C.P.M. Tak, Dennis O. Mook-Kanamori, Hedwig M.M. Vos, Mattijs E. Numans, Niels H. Chavannes

**Affiliations:** 1Department of Public Health and Primary Care, Medical Center, Leiden University, Hippocratespad 21, 2333 ZD Leiden, The Netherlands; d.o.mook@lumc.nl (D.O.M.-K.); h.m.m.vos@lumc.nl (H.M.M.V.); M.E.Numans@lumc.nl (M.E.N.); 2Medical Center-Campus The Hague, Leiden University, Turfmarkt 99, 2511 DC The Hague, The Netherlands; n.h.chavannes@lumc.nl; 3Department of Education, Culture and Wellbeing, Municipality of The Hague, 2542 ED The Hague, The Netherlands; erwintak2@gmail.com; 4Department of Clinical Epidemiology, Medical Center, Leiden University, Postbus 9600, 2300 RC Leiden, The Netherlands; 5National e-Health Living Lab, PO-Box 9600, 2300 RC Leiden, The Netherlands

**Keywords:** older citizens, ageing, technology, digital

## Abstract

In the iZi study in The Hague, use and acceptance of commercially available technology by home-dwelling older citizens was studied, by comparing self-efficacy and perceived physical and mental Quality of Life (QoL)-related parameters on an intervention location of 279 households and a control location of 301 households. Technology adoption was clinically significantly associated with increased perceived physical QoL, as compared with control group, depending on the number of technology interventions that were used. A higher number of adopted technologies was associated with a stronger effect on perceived QoL. We tried to establish a way to measure clinical significance by using mixed methods, combining quantitative and qualitative evaluation and feeding results and feedback of participants directly back into our intervention. In general, this research is promising, since it shows that successful and effective adoption of technology by older people is feasible with commercially available products amongst home-dwelling older citizens. We think this way of working provides a better integration of scientific methods and clinical usability but demands a lot of communication and patience of researchers, citizens, and policymakers. A change in policy on how to target people for this kind of intervention might be warranted.

## 1. Introduction

Increased life expectancy is posing a challenge on economies in general and the healthcare system because not all years gained are healthy years [1]. The increase of our ageing society is a positive yet challenging phenomenon, as population ageing, and urbanization are the culmination of successful human development [2]. The interaction of ageing and urbanism, which is termed urban ageing [3,4], raises issues for all types of communities in various domains of urban living [5]. The demographics of a society with an increasing number of older citizens asks for smart solutions to maintain quality of life and quality of care [6]. Policymakers and technology producers state that technology will help in providing these solutions. Evidence of effectivity of technology is present [7], however pilots are usually performed in closed communities or by means of single-platform tools. Solid proof of effects of technology in this field was currently lacking. In 2010, a review of reviews led to the conclusion that telemedicine is effective, but that evidence in a lot of studies is promising but incomplete and in a lot of other studies is limited and inconsistent. Major problems were found in economic analyses, benefit for patients, complexity of telemedicine, and ongoing collaborative achievement in unpredictable processes [8]. This inhibits investments of public and private parties in development and implementation of technology.

In the Netherlands, a transition has been made to shift the care for older citizens from centrally organized to municipally organized. The aim is to reduce formal and intramural care to promote healthy home-based aging. This necessitates another way of involvement of both formal and informal caretakers.

In the municipality of The Hague, the “iZi study”, focusing on technology adoption by older citizens, aimed to experiment with introducing technology in a need-driven way. This provided opportunities to tailor those needs and connect them to possible technologic solutions. In this study, the municipality of The Hague has taken the initiative to experiment with technology introduction to establish how it can contribute to the transition soon.

In this study, we aim to evaluate whether need-driven introduction of technology in adult humans leads to improved self-reliance and a better perceived quality of life. We think that need-driven introduction and use of technology will lead to positive effects on aging in place.

## 2. Materials and Methods

### 2.1. Study Design

The iZi pilot is a prognostic controlled observational study and was performed in a community in the municipality of The Hague. The community consists of community-dwelling older citizens (55+). The total complex counted 279 households. Because the intervention was tailored to individual needs, and therefore differed for each participant, the intervention was evaluated based on intention-to-treat analysis. The participants on the iZi location were compared with citizens on a location in The Hague that was matched on relevant characteristics, to be comparable to the intervention location. This community consisted of four apartment buildings, with a total of 301 households.

The process of inclusion is depicted in a flow chart that is attached as Figure A1.

We have chosen to evaluate effects in this study by comparing the differences in effect between T0 and T12 of participants in the study with the control population.

On the intervention location, residents were matched to technology from a preselected list, based on previous needs assessment and selected together with participants in group sessions. All items were supplied for the pilot by technology producers and suppliers. This list is attached as Table A1.

On the intervention location, participants were included in a stepwise fashion. First, they were recruited by using several strategies such as group meetings, door-to-door-calls by the iZi team, and posters and messages on the (specifically installed) electronic bulletin board. Once residents participated in the technology trial, they were approached to participate in the effect evaluation by giving informed consent. Participants who had given informed consent for the evaluation were visited by a research nurse for evaluation at T = 0 and T = 12 months.

On the control location, participants were recruited by letters sent by the housing corporation. They could respond by sending an enclosed response form to a free mail address but were also provided with a telephone number and an e-mail address. After a resident responded, the research nurse called to provide information, and if people chose to participate, an appointment was made in which informed consent was obtained and evaluation at T = 0 and later T = 12 was performed.

The baseline characteristics of the intervention population and the control population are described in Table 1.

During the study, the inclusion targets for the effect analysis were not met. Differences between two groups were seen in the composition of the households of the participants. On the intervention location, a higher number of singles participated than on the control location.

### 2.2. Self-Reliance and Participation

Self-reliance was measured using the “Impact on Participation and Autonomy Questionnaire” (IPAQ) [9,10]. The IPAQ consists of five domains: autonomy indoors, family role, autonomy outdoors, social relations, and work and educational opportunities. Each domain consists of a different number of items. Answers were scored on three- and five-point Likert scales. For each domain, the participation score and the problem experience score were calculated by summing the item scores. Higher scores denote more restrictions in participation and/or a higher problem experienced on the specific domain.

### 2.3. Quality of Life

Quality of life (QoL) was scored using the SF-12 health survey, a survey consisting of 12 items in which physical functioning, role limitations due to physical health, bodily pain, general health perceptions, vitality, social functioning, role limitations due to emotional problems, and mental health are scored [11]. The SF-12 scores were calculated using the general scoring system provided by the developers [12]. Since country-specific scoring did not differ much [13], the original scoring system was used to provide better external validity.

In the general population, the average score is set on 50, with 0 representing the worst possible health and 100 representing the best possible health. In the Dutch population, the physical component score on SF-12 tends to diminish with increasing age. The mental component score shows no such tendency. The mean score of Dutch women on both components is lower than that of men. This effect is consistent in all age groups [13].

### 2.4. Establishing Statistical and Clinical Significance

To get significant results, a certain number of participants are needed. Therefore, a power calculation was performed, using the IPA norm scores [14].

To detect statistically significant changes at a 5% significance level with 80% power, with two-sided testing, inclusion of at least 112 households on the intervention location and 120 households on the control location would be necessary to be able to detect a 10% change.

For clinical significance of measured effects, we could not do prior calculations of number of participants needed, however we call effects clinically significant if a change occurs that has consequences in the real-life world of participants. This practically means that the effect is seen after a relatively short amount of time and has a magnitude that is perceived either directly by the participants or can be picked up by the researchers by means of their measuring tools.

### 2.5. Statistical Analysis

For some participants, scores could not be calculated due to missing data. Linear regression analysis was used to establish whether possibly significant effects due to differences in age, gender, and marital status were missed. We compared the differences in the mean scores using paired *t*-test statistics in SPSS statistics 25 (IBM, New York, NY, USA).

First, we compared the differences in IPAQ score between T0 and T12 in both the iZi group and the control group separately, using paired *t*-tests. For this analysis, we examined the five main dimensions of the IPAQ score. Next, we compared the differences of the two scores (mental and physical) of the SF-12 between T0 and T12 in both the iZi group and the control group separately, using paired *t*-tests as well. The outcomes for both populations are described in Tables 3 and 4. Linear regression analysis, performed to establish whether possibly significant effects due to age, gender, and marital status were missed, yielded no significant effects.

After that, we subtracted the score of T0 from the score on T12 for each specific person on each specific location, thus creating the specific change on a dimension (delta) for each person on each location. We compared the median value of this delta for both locations, using the Mann−Whitney U for testing on significance.

Finally, we checked whether the significant changes that also had a clinical significance could be ascribed to the amount of technology offered. We also explored whether a specific kind of technology could be related to the statistical or clinical effect seen.

## 3. Results

The results of both groups at both timepoints on the different domains of the IPAQ are depicted in Table 2.

The results of both groups at both timepoints on the different domains of the SF-12 are depicted in Table 3.

Evaluation of IPAQ yielded statistical significant difference in the scores for autonomy indoors in the intervention group and social relations in the intervention group. However, the magnitude of the effect was so little that we did not see this as a workable starting point to perform further evaluation of these domains into subgroups.

Evaluation of SF-12 yielded statistical significant differences in the scores for physical health in the control group and statistical significant differences in the scores for mental health for both the control group and the intervention group.

We then compared the differences between T12 and T0 of both intervention and control group, to evaluate whether the differences observed could be related to the intervention. (Subtracting measurements of T0 does provide a “net” value that makes comparison possible if the groups are equal in composition).

The results of this comparison are depicted in Table 4.

We assessed whether the effect on perceived physical health might be related to intensity of technology use. On the iZi location, participants were matched to between zero and seven different technologies. Distribution of technology is described in Table 5.

Because few participants had four or more technologies, for further analysis a division was made into four groups: participants with zero, one, two, and three or more technology matches. For practical reasons, we put all the people who did participate in the evaluation but did not get (control location) or did not get matched to technology (intervention location) in the “no technology” group.

This is depicted in Figure 1.

What is seen is a decrease in perceived physical QoL when persons go from no technology to starting technology, but from there an almost linear trend of increasing QoL with increasing number of technologies can be seen.

Because a lot of different technologies were used, we performed a subgroup analysis based on two different classifying systems: a classification on technological level and a functional classification. We checked all classifications for subclasses that revealed useable patterns.

The classification of technological level was made in three groups: high-tech, low-tech, and in-house mechanical adaptations. The division high tech vs. low tech was made from the apparatus and user point of view, labelling high tech all those technological interventions that had complex operating systems or provided multiple functions or options.

The functional classification, with which the intervention originally started, divided technology into groups depending on the domain of life to which they were attributed and matched, and used the following classes: safety, mobility, communication. The classifications and the used technologies are depicted in Table A1.

We visually evaluated graphs for relevant results. Only the classification on technological level yielded possible relevant differences. These differences are depicted in Figure 2, Figure 3 and Figure 4.

In our follow-up, being matched to technology was only associated with an increase in perceived physical health in the group that received in-house adaptations. In the high-tech and low-tech groups, there was a decrease in perceived physical health.

## 4. Discussion

The goal of all long-term care arrangements is to reduce the disabling effects of physical impairments and functional limitations. However, the means with which individuals cope with disability may not be equivalent and these differences may influence self-reports of disability in surveys. Agree et al. [15] examined assistive devices and personal care as factors in the measurement of disability among persons aged 70 and concluded that the use of assistive technology differs from personal care on a fundamental level. They also concluded that it does not require the ongoing cooperation or coordination of other people and therefore increases the sense of independence with which a disabled individual can meet their long-term care needs. Their results indicate that older individuals can expect to spend most of their remaining years in good functional health, but up to two-thirds of disabled years will be spent with unmet needs. Among those who are disabled, those who use only equipment and no personal care report less residual difficulty with mobility than those who use personal assistance (either alone or in combination with equipment), but the use of equipment alone is most effective for those with the least severe limitations. This coincides with the finding in our study that in-house adaptations provide the largest shift in QoL.

A lot of technology is available for aging societies, but acceptance amongst older citizens is difficult [16], and there are also ethical dilemma’s involved [17]. Technology introduction for older citizens is difficult, because the ability to learn new things usually wanes with age and might even be worse in those who might potentially benefit the most. However, effectiveness of technology in people with learning abilities has been shown [18]. Tailoring interventions to personal needs might help to improve acceptance, adoption, and use of technology [9]. Thus introduced, technology might enable older citizens to age at home [19]. Our study shows that in tailoring interventions, older citizens are accepting and adopting technology. The use of technology, be it high-tech or low-tech, is associated with a decrease in QoL. The effects on physical health seem in this study only to be associated with in-house adaptations and not with high-tech or low-tech solutions. However, the follow-up period was quite short, and a relative low number of high-tech and low-tech solutions were matched. What possibly might influence the results is the fact that in the prior consultation, most people rated accessibility as one of the most important domains. Therefore, most people have been matched to items in this range, and those were mostly low-technology items. The needs that require high-technology solutions are fewer and therefore do not contribute to a large component in the effect, but positive results are seen by researchers and reported by participants that have been exposed to high-technology solutions.

It turns out that technology might help in facilitating adaptive changes, but that implementing the right technology at the right moment for the right reason and the right person is quite difficult. Currently, tools are being developed to support this process [16]. What tool is suited for which problem is difficult to ascertain. Efforts have been made to devise a taxonomy describing essential features of interventions [13].

Most people prefer to age the way they have lived all their life, however, sometimes circumstances change. These changes in circumstances may be the result of a process that has been going on for years, but may also be provoked by an event (for example changes in living arrangements or illness). These changed circumstances may also affect the relationship with the informal caretaker (usually the partner, child, or other family member) and may disrupt the status quo, necessitating adaptive changes. It is important to identify people that may be prone to benefit from technology when these changes are to occur, to assess their needs, and to see if those needs can be met [9]. Since the needs within this population differ, communication about needs is vital.

The iZi study in The Hague shows that, contrary to earlier studies and beliefs amongst caretakers, older citizens are interested in the adoption of technology and once recruited and enabled to partake in the matching process, can successfully adopt technology. Technology adoption is associated with an increased perceived physical health as compared with the control group, depending on the number of technology interventions that are used.

Whether this is due to the technology or whether participants in this study were healthier at the start of the study is something that remains to be established. In general, this research clearly shows that successful and effective adoption of technology by older citizens is feasible with commercially available products amongst home-dwelling older citizens.

Unfortunately, from a point of view of cost-effectiveness the intervention cannot be implemented in the way it was researched on a larger scale, since on the intervention location a lot of people were involved in the matching process. From a funding point of view, cost-effectiveness can be evaluated by looking at it from both municipal and health insurance budgets, because at this point municipalities must pay for the intervention, while the gain is made on the health insurance budgets. Unfortunately, health insurers were not involved in this study.

During the intervention, the idea arose that the most vulnerable citizens were possibly not targeted by this specific recruitment. The intervention managed to reach only a part of the older citizens, and from analysis of subgroups that we did, we had the idea that especially older citizens from other cultures and people dealing with social issues or diseases tended not to participate. This leaves the question open whether, apart from the fact that this intervention shows a positive change in the intervention group as compared with the control group, this intervention (if there is a cause−effect relation) might have the potential to have an effect if it would access populations that could benefit from it in a better way. Therefore, it would be interesting to establish which older citizens stand to gain the most from which interventions, and how the matching and recruitment process can be improved in the future. Collaboration between GPs and municipality might be helpful in selecting and reaching out to older citizens who might benefit. More research on this subject is needed to establish if this indeed is a successful route.

The statistical significant increase in perceived autonomy indoors and social relations compared with the control population has a magnitude and therefore a clinical significance that is small. These results might be due to a lack of power, but it might also suggest that even if people want to use technology, getting used to using technology requires an effort and might also pose difficulties that are not so easily overcome [16]. It would be interesting to know what the effects are over a longer period, since a lot of time and effort was spent on getting used to the technology. The time lost on solving functionality issues and testing might have negatively influenced the perceived physical health, because getting to work with technology that helps solve physical issues requires that people do recognize and accept those issues. So paradoxically, for technology to be able to help people function in a better way, it might start by making them feel worse. If this is indeed the case, it is important to recognize and describe this process, to prevent premature abortion of potential successful interventions.

The statistically and clinically significant decrease in perceived mental health in both populations warrants further exploration.

Moreover, difficulty in reaching older citizens who might really benefit and difficulty in obtaining technology that was really tailored to the needs of these citizens might have given an underestimation of the real effect of technology. The initial results show that successful and effective adoption of technology by older citizens is feasible with commercially available products. The iZi project established a way to measure clinical significance by using mixed methods, combining quantitative and qualitative evaluation and feeding results and feedback of participants directly back into our intervention. We think this way of working provides a better integration of scientific methods and clinical usability but demands a lot of communication and patience of researchers, citizens, and policymakers. Given our results, this warrants further exploration. Especially, more research is needed to establish the existence of unmet care needs amongst this specific population to evaluate if the effect we see can be improved, thus making the intervention—if there is a causal relation between intervention and effect—possibly more cost-effective.

## 5. Conclusions

Though target inclusions were not met in this study, we were able to demonstrate a decrease in perceived physical health in the control population that was not seen in the intervention population. Both populations showed a decrease in perceived mental health, measured with the SF-12. Lastly, on the intervention location, an increase in perceived autonomy indoors and social relations compared with the control population was measured. The clinical effect of this change was however small.

An increase in the number of technology matches was associated with an increase in perceived physical health with three or more successful matches to technology. In our subgroup analysis, being matched to technology was only associated with an increase in perceived physical health in the group that received in-house adaptations. In the high-tech and low-tech groups, there was a decrease in perceived physical health.

## Figures and Tables

**Figure 1 ijerph-17-05052-f001:**
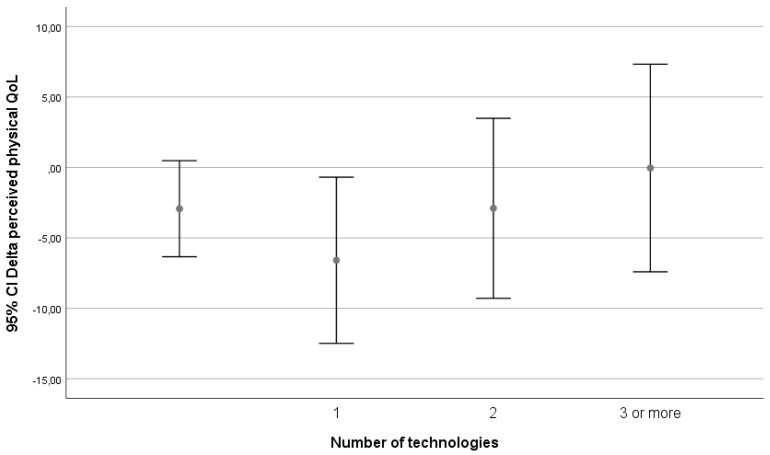
Relation between delta perceived physical health (SF-12 _(physical health T12)_ − SF-12 _(physical health T0)_) and number of technologies (1,2 or 3 or more) compared with those who did not get or did not get matched to technology.

**Figure 2 ijerph-17-05052-f002:**
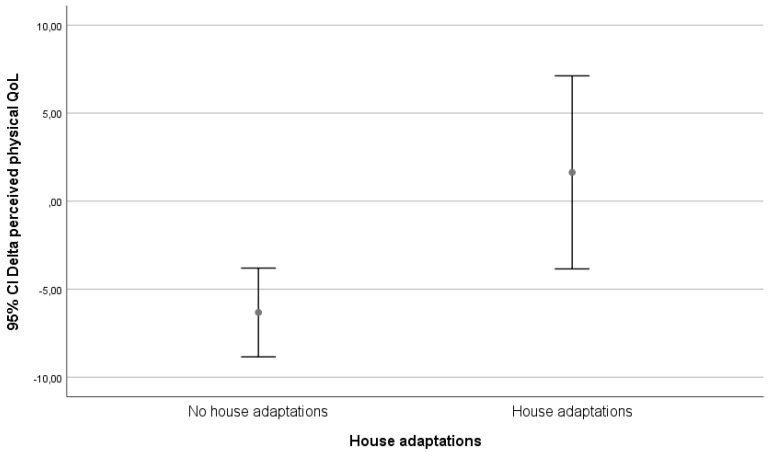
Relation between delta perceived physical health (SF-12 _(physical health T12)_ − SF-12 _(physical health T0)_) and match to in-house adaptations.

**Figure 3 ijerph-17-05052-f003:**
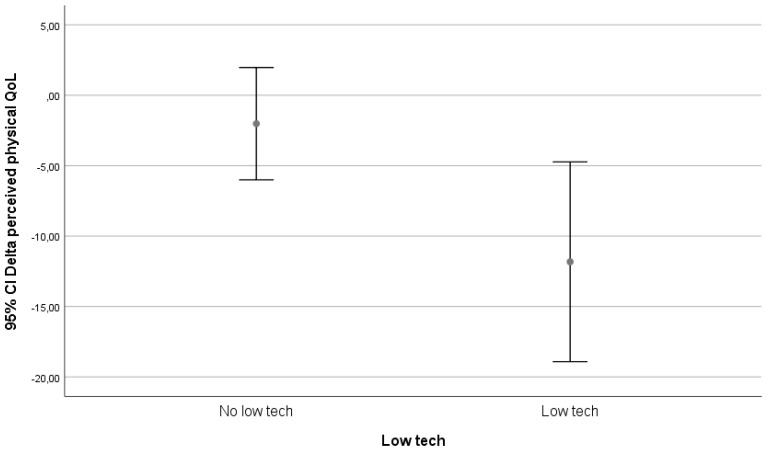
Relation between delta perceived physical health (SF-12 _(physical health T12)_ – SF-12 _(physical health T0)_) and match to low-tech technology.

**Figure 4 ijerph-17-05052-f004:**
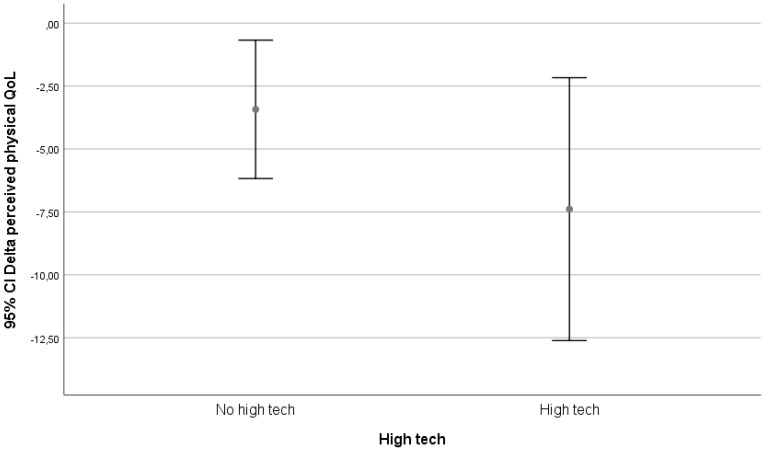
Relation between delta perceived physical health (SF-12 _(physical health T12)_ – SF-12 _(physical health T0)_) and match to high-tech technology.

**Table 1 ijerph-17-05052-t001:** Composition of intervention and control group at different timepoints.

Composition	Intervention T0	Intervention T12	Control T0	Control T12
*n*	83	56	77	48
Age	(%)	(%)	(%)	(%)
51–65	8.4	5.4	11.7	8.3
66–70	10.8	12.5	13.0	14.6
71–75	25.3	25.0	18.2	20.8
76–80	30.1	30.4	27.3	29.2
>81	25.3	26.8	29.9	27.1
Sex	(%)	(%)	(%)	(%)
Male	39.8	41.1	35.1	33.3
Female	60.2	58.9	64.9	66.7
Marital status	(%)	(%)	(%)	(%)
Living together	37.3	32.1	53.2	45.8
Single	62.7	67.9	46.8	54.2

**Table 2 ijerph-17-05052-t002:** Scores of intervention and control group on IPAQ at different timepoints.

IPAQ Domain	Intervention T0	Intervention T12		ControlT0	Control T12	*p*
	Mean (SD) *n*	Mean (SD) *n*	*p*	Mean (SD) *n*	Mean (SD) *n*	
Autonomy indoors	0.97 (0.66) 76	0.98 (0.51) 48	0.02	0.97 (0.57) 82	1.10 (0.28) 56	0.24
Family role	1.32 (0.73) 73	1.19 (0.46) 47	0.60	1.34 (1.01) 78	1.30 (0.42) 56	0.68
Autonomy outdoors	1.37 (0.79) 74	1.17 (0.44) 47	0.90	1.30 (0.77) 79	0.32 (0.36) 56	0.84
Social relations	1.11 (0.58) 71	1.10 (0.32) 43	0.05	1.07 (0.59) 78	1.19 (0.18) 55	0.67

*p*-values calculated with paired *t*-test.

**Table 3 ijerph-17-05052-t003:** Scores of intervention and control group on SF-12 at different timepoints.

SF-12 Domain	Intervention T0	Intervention T12	*p*	Control T0	Control T12	*p*
	Mean (SD) *n* = 79	Mean (SD) *n* = 46		Mean (SD) *n* = 74	Mean (SD) *n* = 56	
Physical health	41.42 (12.15)	39.90 (4.81)	0.57	41.95 (11.61)	38.69 (6.21)	0.002
Mental health	43.42 (9.85)	39.78 (6.33)	<0.0001 *	43.72 (9,71)	42.29 (6,60)	<0.0001 *

*p*-values calculated with paired *t*-test. * Statistical and clinical significant.

**Table 4 ijerph-17-05052-t004:** Comparison of differences in health scores of intervention group and control group.

IPAQ Domain	Intervention	Control	*p*−Value
	Delta Median	Delta Median	
	(Interquartile Range)	(Interquartile Range)	
Autonomy indoors	0.00 (−0.23, 0.75)	0.00 (−0.14, 0.57)	0.649
Family role	0.00 (−0.14, 0.29)	0.00 (−0.36, 0.50)	0.380
Autonomy outdoors	0.00 (−0.20, 0.40)	0.20 (−0.40, 0.55)	0.975
Social relations	0.00 (−0.29, 0.75)	0.07 (−0.14, 0.43)	0.315
Physical health	−7.85 (−12.70, 4.08)	−2.42 (−13.18, 7.91)	0.037
Mental health	−1.35 (−9.94, 9.03)	−4.79 (−9.65, 2.93)	0.124

*p*-calculated using Mann-Whitney U.

**Table 5 ijerph-17-05052-t005:** Number of technologies matched to a household.

Number of Technologies	Number of Households	% of Households
0	203	69,0
1	30	10,2
2	28	9,5
3	19	6,4
4	8	2,7
5	4	1,4
6	1	0,3
7	1	0,3
Total	294	100,0

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
