# Peer review of "Effects of Technology Use on Ageing in Place: The iZi Pilots"

_ijerph, 2020, doi:10.3390/ijerph17145052_

Round 1

Reviewer 1 Report

First remark: The authors have compared two groups before and after an intervention so a comparison of 4 means (intra group before and after 12 months and inter group before and after 12 months). So, the statistical analysis must be a two-way repeated measures analysis of variance: ‘‘groups’’ (EG vs CG) versus ‘‘time’’ (0-12 months) factors. What is the level of significance accepted?

Second remark: the number of comorbidities per group is not known which makes any hazardous conclusion in relationship with the aim of the study; Line 82 : what are the “relevant characteristics of the subjects”, for example the level of education can induce some facilities to accept/learn new technologies; from an experimental point of view, these remarks are a fatal flow.

Third remark: the authors must question about the degree of drop out and discuss this point in the part discussion; at the beginning of the recruitment, the number of the subjects for the experimental group was 294, with some explanations about the aim of the research, the number was fallen at 115 and at the end of the experience, there are just 56 subjects. We can us ask the interest of the new technologies among all people with have been drop out. The same remark can apply to the control group.

Fourth remark, line 144 and 146: dividing three groups by an intuitive methodology is still an absence of scientific reasoning.  

Part abstract: the authors must be nuanced the result concerning the sentence: “a higher number of adopted technologies is associated with a stronger effect”, the real results demonstrated this effect only for physical QoL moreover associated with the error experimental flaw designed above.

Reviewer 2 Report

Referee report on “Effects of technology use on ageing in place: the iZi pilots”

Brief Preview:

This paper attempts at studying the effects of usage of technology at home by older citizens. The authors do so by tracking several parameters and for 301 households. The authors show that aging people can effectively adopt technology at home due to its availability.

The paper’s idea of linking technology and the aging population is good, however, the paper requires major amendments for it to be published.

Abstract:

  • If the paper ends up by policy recommendations, then better mention that in the abstract
  • No need to mention that we need more research in the abstract. This may be mentioned in detail at the end of the paper in the conclusion section.
  • Readers shall know implicitly whether what you are doing is “promising”. One should refrain from evaluating his work!

Introduction:

  • The authors started by motivating for their work however I can’t find the link between the presented motivation and the research question.
  • What is the contribution of the paper? For example, does it aim at filling a certain gap in the literature?
  • The authors opted to include one section that serves as an introduction and a literature review all at once. Although this is technique is acceptable, one would expect a lengthy introduction under such a scenario, rather than a one page worth of introduction and literature review combined! This may be an indication of poor research into the existing body of work, which in turn raises doubt about the full understanding of the research being conducted by the authors within the framework of previous research on the subject matter.

Having said that, authors are highly advised to consider the below papers among other sources of their own:

  • Fakih, A. (2014). Vacation leave, work hours, and wages: New evidence from linked employer–employee data. Labour28(4), 376-398.
  • Czaja, S., & Schulz, R. (2006). Innovations in technology and aging introduction. Generations30(2), 6-8.

Materials and Methods:

  • It is recommended that the authors merge the “study design” section with any other section under the materials and methods as there is no need for a separate subsection just for 4 lines.
  • The authors mentioned that “The effects of aging differ greatly”. How do they differ? Is this backed up by any literature? If so, authors may want to elaborate more on this subject.
  • The section titled “Setting and participants” reads like a description of how data was collected. Readers are not likely to be interested in the detailed explanation of the data collection process rather the main findings, the contribution and the policy recommendation generating from the research work.

However, if the authors believe – for any reason - that this section is important to be mentioned, then they may want to move it to an Appendix at the end of the paper to leave room for the things readers are more interested in.

Results:

  • The results section is brief! The authors are strongly advised to revisit this section and see how they can improve it. A viable way is to merge the discussion section with the results section. Authors are also advised to delve deeper into the existing body of literature to get a better perspective of the research being done on the subject matter. This may reveal unique findings of the paper as opposed to the findings of the literature.
  • Additionally, graphs should be moved to the end of the paper to avoid having a messy paper.

Conclusion

Similar issue with the results section, the conclusion section is brief and lacks relevant policy implications. What is the benefit of the established work? Authors would like to spend some time answering this question here

Minor comments:

  • It is enough to include three to five keywords rather than ten
  • Sentence Structure: the structure of sentences, in general, require major revision to ensure clarity.

Overall Assessment:

This paper provides some good approach however it requires major amendments shall it be accepted for publication later on.

Round 2

Reviewer 1 Report

i agree with the responses of the authors

Reviewer 2 Report

The paper is improved and publishable.